# Policy Evaluation and Policy Style Analysis of Ride-Hailing in China from the Perspective of Policy Instruments: The Introduction of a TOE Three-Dimensional Framework

Xintao Li [1,2], Shuochen Zhang [1,*], Diyi Liu [1], Tongshun Cheng [1,2] and Zaisheng Zhang [3]

1 Zhou Enlai School of Government, Nankai University, Tianjin 300350, China
2 Chinese Government and Policy Joint Research Center, Nankai University, Tianjin 300350, China
3 College of Management and Economics, Tianjin University, Tianjin 300072, China
* Correspondence: zhangshuochen@mail.nankai.edu.cn

**Abstract:** Online ride-hailing in China brings convenience for the public, but it has caused several problems, such as inadequate supervision, data security risks, and financial risks. This new industry has also disrupted the traditional taxi market. China's government implemented some policies, which were initially disorderly tightening, and then formed the policy system responding to various needs for tackling these issues gradually. There were some policy fluctuations and regulatory effects during this period, therefore, it is imminent to evaluate the online ride-hailing policy text. In this paper, we took 43 online ride-hailing policies as samples, with the consideration of policy instruments and statistical inspection methods. In this paper, we also constructed an innovative three-dimensional analysis framework by combining content analysis, and further identify the ride-hailing policy development during different stages of development periods (2016–2022). Digging into the problems existing in the new online ride-hailing, policies were drawn by module division, unit coding, inductive statistics, the quantitative evaluation of policy text content, and TOE (technology-organization-environment) style analysis. Finally, we provide insightful policy recommendations for online ride-hailing policies, committed to providing theoretical support and a decision-making basis for governance policies in the transportation industry.

**Keywords:** online ride-hailing policy regulation; policy instruments; TOE framework; policy style

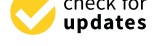



## 1. Introduction

With the development of the sharing economy, information technology, and globalization, the growing pattern of online ride-hailing, represented by Uber, has quickly found a suitable growth environment in China, which has had a great impact on China's traditional taxi industry. Therefore, the symbiosis concept of the sharing economy offers an effective operating mechanism for the duality of innovative market economic methods and social goals [1]. Furthermore, the Zero Marginal Costs Society of the sharing economy provides effective micro-mechanisms for new types of organizations, such as hybrid organizations [2]. However, as an important part of the sharing economy, the gig economy will also present obstacles to social progress, for instance: the damage to workers' rights and interests; the deterioration of social security; and market disruption. In Hustle and Gig, Alexandra J. Ravenelle provides sociological insight into the impact of the sharing economy on ordinary citizens, and argues that this gig economy, featuring flexible and win-win work models, is essentially a movement forward to the past [3].

According to statistics published by China's Online Ride-hailing Regulatory Information Interaction Platform, as of 30 April 2022, a total of 270 online ride-hailing platform companies in China were granted business licenses to operate online ride-hailing platforms. At the same time, 4.229 million driving licenses and 1.754 million vehicle transport licenses were issued in various regions. Since 2010, the online ride-hailing market has continued to

expand, with Didi's daily orders exceeding 10 million as of 19 March 2016. Furthermore, the Didi now operates in more than 400 cities [4]. Two years after its launch in 2012, the online car-hailing platform had successfully attracted a large number of consumers and drivers through advertising, price cuts, and subsidies. This has directly led to market competition between online car-hailing companies and traditional taxi operators, while Internet companies have greatly affected the development of cruise taxis and related industries by taking advantage of information and price advantages. This situation has led to a conflict of interest between online ride-hailing drivers and cruise taxi drivers, manifested as the outbreak of mass incidents in some cities (In 2015, Changchun, Jinan, Chengdu, Nanchang, and other cities, information source: https://news.sina.com.cn/c/p/2015-01-13/193631394903.shtml (accessed on 3 June 2022). At the same time, the operation mode of this industry is to attach private cars to leasing companies, hire drivers through labor dispatch companies, and sign "four-party agreements. The "four-party agreement" model is an agreement signed by the online car platform company, the car leasing company, the labor dispatch company, and the driver, to regulate each other's rights and obligations. This type of cooperation bypasses the government's regulatory policy on the traditional taxi industry and tests the government's ability to regulate. Negative reports about price discrimination, sexual assault, and uncivilized driver behavior in the online ride-hailing market have all seriously affected the user's perception of service quality and corporate reputation [5], throwing the industry into market chaos. Therefore, to prevent the uncontrolled spread of negative emotions and to ensure the good operation of the market, governments have changed their style of management from laissez-faire development to strict control [6]. Due to a series of complex factors, such as innovation demand, high-level promotion, and market incentives, the government has gradually eased restrictions in the online ride-hailing industry.

Through observation, it is found that the online ride-hailing policy of China has evolved in line with the general economic and social environment, and the development status of the online ride-hailing industry. In October 2015, the Ministry of Transport formulated the "Interim Measures for the Management of Online Taxi Booking Services (Draft)", revealing the negative attitude of the central government towards online ride-hailing services. However, on 26 July 2016, the policy was relaxed, to some extent. The General Office of the State Council issued the "Guiding Opinions of the General Office of the State Council on Deepening Reform and Promoting the Healthy Development of the Taxi Industry", which laid the basic framework for the management of the online ride-hailing industry. The following day, seven ministries and commissions, including the Ministry of Transport and the Ministry of Industry and Information Technology, jointly issued the "Interim Measures for the Management of Online Taxi Reservation Services", which aimed to further standardize online car-hailing operations and service behavior, safeguard the industry environment, and protect the legitimate rights and interests of passengers. On 5 June 2018, the Ministry of Transport and seven other ministries and commissions jointly issued the "Notice on Strengthening the Joint Supervision of the Online Taxi Reservation Industry", which required the establishment of a joint supervision mechanism for all departments at all levels. Overall, China's ride-hailing policy has gone through several different stages of development, showing different styles of regulatory policies. At the same time, the institutional logic behind it is rooted in various conditions of economic and social development, and reflects the result of the joint action of various factors.

These policies have gone through several different stages of development, presenting regulatory policies with different styles. However, policy-level research on the online ride-hailing industry has not received commensurate academic attention, with some notable exceptions, by empirically examining the acceptance of new policies by online ride-hailing drivers in Shenzhen, China [7]. In particular, there is still a lack of scientific and systematic answers to the style and policy evaluation of China's central-level online ride-hailing policies.

This study bridges these research gaps by systematically examining the development process of China's central-level online ride-hailing policies, and introducing a TOE framework to analyze the style of China's online ride-hailing policies based on policy

classification by using policy instruments. There are two important reasons for using policy instruments for classifying these policies. First, according to the principle of moderation, things undergo an accumulation of quantitative changes before they undergo qualitative changes. The development of online car policies is also a process of quantitative to qualitative change that occurs under the influence of the external environment and internal factors. Therefore, the analysis of the style of online ride-hailing policy needs to include the dimension of time. In addition, policy formulation and implementation are directly influenced by policy objectives, where the policy instrument classification method plays a key role in the process of assessing the desired objectives. Therefore, this paper introduces the TOE framework to systematically examine the stylistic changes in China's central-level online ride-hailing policies. The combination of this framework and policy instruments provides a new theoretical perspective for policy analysis research, as it introduces the influence of environmental variables on policy based on the original classification of policy instruments considering internal factors, which can reflect policy styles in a three-dimensional way. Afterward, based on the view of the policy instruments and statistical test, online ride-hailing policies in recent years were used as samples; the three-dimension analysis framework, by using the content analysis method, was established and module division, unit coding, induction and statistics, and quantitative evaluation, were applied. Then, we analyzed the online ride-hailing policy style in China and dug into the problems existing from an analytical framework constructed from three dimensions: policy instruments, policy developments, and the TOE framework. Finally, some policy recommendations were proposed based on the issues and policy styles that were previously evaluated.

## 2. Literature Review

Online ride-hailing (e.g., Uber, Lyft, Didi, etc.), also known as real-time ride-sharing, ride-hailing, on-demand rides, and ride-sourcing [8], is an emerging service form of the sharing economy. In the process of development, it is regulated and constrained by different types of supervision and management, or development promotion policies of governments around the world, including setting market access thresholds, controlling price fluctuation ranges, and regulating vehicle models and driver assessment. Therefore, scholars have analyzed the utility of online ride-hailing policies in various countries.

### 2.1. Ride-Hailing Policy

The content analysis of the car-hailing policy text contains the intention of the policy implementer, so it has practical significance. The current research on car-hailing policy mainly includes the necessary demonstration of the regulatory policy, the evaluation of the effect of the policy implementation, and the influencing factors of the policy introduction. As there are potential risks, such as privacy risks and security risks in ride-sharing, the user's willingness to purchase carpooling services has been affected as a result [9]. Some scholars analyzed seven factors that affect the passenger's perception of safety and ride experience through empirical research, and further argued that the government must introduce some regulatory measures and policies to improve the safety of online ride-hailing [10]. For example, two cases in China in 2018, in which female passengers were viciously killed by drivers, while carpooling through the Didi platform, put the carpooling service into a safety crisis. In the wake of this, the central government has increased attention on the regulation of online ride-hailing and has developed a series of new regulatory policies. Tamer Cetin and Elizabeth Deakin argued that: "entry and fare regulations are not efficient policy measures to solve market failures in the taxicab market" [11]. But some scholars hold the opposite view on this issue. An empirical study in 2019 divided the risks associated with ride-hailing services into institutional, economic, and safety aspects, and the empirical results showed that public policies have a significant impact on reducing the conflicts and risks of shared mobility in China, especially institutional risks [12]. It can be seen that there are significant differences in cross-regional and cross-policy action areas in the evaluation results of online car-hailing policies. One study further analyzed the factors that influence

the effectiveness of policy implementation, concluding that the utility of the policy text will also be affected by the way the policies are promulgated, as well as environmental factors [13]. When examining these factors, researchers often compare the content and effectiveness of online ride-hailing policies between the central and local governments of different regions. It has been argued that under a centralization of authority, the central government has more comprehensive and strict supervision of the platform, while there are significant differences in policies between different local governments, and policies that are issued later tend to be stricter [12]. Sharif Naubahar and Xing Jack Linzhou compared the online ride-hailing policies formulated by Xi'an, Chengdu, Beijing, and Guangzhou, and found that different local economic development environments, urban strategic positioning, the uniqueness of environmental protection, and other factors associated with large differences in policymakers' attitudes towards online ride-hailing, which has led to the formation of diversified local policy innovations [14].

### 2.2. The Use of Policy Instruments in Policy Analysis

As a key way of understanding the measures taken by government organizations to achieve specific economic or social objectives, the policy instruments are not only the central element of the policy text, but can also help us to classify the policy system and evaluate the policies from multiple dimensions [15]. Currently, the policy instrument is widely used in the construction of policy systems in various fields, such as environmental protection [16–18], education [19,20], transportation [21], and diplomacy [22,23]. These methods have widespread applicability and are conducive to analyzing China's online ride-hailing policy.

There is often no fixed way of classifying policy instruments, and researchers mostly used one type of policy instrument that is appropriate to the subject of study to classify policies. Some researchers categorized environmental policy instruments into market-based and command-based instruments, exploring how environmental policies should be designed [24]. Others classified policy instruments that stimulate innovation into supply-push and demand-pull types, and explored which policies can promote innovation in renewable energy technologies [25]. However, recent studies tend to treat policy instruments as a single dimension of analysis when they are used to classify policies. In other words, the classification method using only policy instruments only examines the internal characteristics of the policy but ignores the external factor of the policy implementation environment, and thus is not conducive to a holistic view of policy development. Some researchers applied the technology-organization-environment (the following is abbreviated as "TOE") framework to policy analysis, examining the factors influencing the acceptance of digital transformation policies by advertising companies from three dimensions [26], addressing the problem of focusing too much on the policy itself and neglecting the influence of the environment in the process of policy analysis, inspiring this study.

In summary, previous research on online ride-hailing policies has focused on micro-analysis of the policies themselves, lacking a macro perspective to assess and analyze the overall development and stylistic features of policies. At the same time, how policy instruments are applied in the process of policy categorization also has the potential for further development. Therefore, to remedy the above misgivings, this paper innovatively introduces the TOE framework into the policy analysis of ride-hailing, based on the classification of online ride-hailing policies using policy instruments, so as to form a holistic understanding of the policy style of online ride-hailing in China.

This paper first reviews academic research on online ride-hailing policy and relevant policy evaluation. A total of 43 online car-hailing policies issued by the central government in recent years were selected as samples for the analysis. A content analysis method was employed to innovatively construct the TOE-policy instrument within a three-dimensional analytic framework, and to conduct module division, unit coding, inductive statistics, quantitative evaluation, and style analysis, of the policy content. Cross-analysis was then carried out to evaluate China's online ride-hailing regulation policy, and to discuss the

policy style reflected therein. Finally, some corresponding countermeasures and suggestions are proposed based on the shortcomings of the current regulatory policies. The potential contributions of this paper are: (1) enriching the scope of the use of policy instruments to analyze methods, and finding a way of classifying policy instruments suitable for analyzing online ride-hailing policies; (2) it expands the analytical dimension of policy instruments and introduces the TOE framework to construct a three-dimensional analytical framework for the analysis of policy styles; and (3) this paper systematically reviews China's central-level online ride-hailing policy, analyzes the development and stylistic changes in these policies, and provides empirical support for online ride-hailing policy reform.

## 3. Research Design and Framework

In 2016, the "Guiding Opinions of the General Office of the State Council on Deepening Reform and Promoting the Healthy Development of the Taxi Industry" clarified the market positioning of online ride-hailing for the first time. This policy also clarifies the basic pattern of the integrated development of online car-hailing and the traditional taxi industry, laying the foundation for subsequent policies. Therefore, this paper takes all the online car-hailing policies and standard documents issued by the central government from 2016 to 2022 (before submission) as the research object. All the basic data were sourced from the official website of the State Council, the Ministry of Transport and the PKULAW Database (https://www.pkulaw.com/, accessed on 25 May 2022). After searching the relevant policy documents, we removed the policy tests with similar content, and selected 43 central policy tests related to online ride-hailing, as shown in Table 1 (The full table is available in detail at Supplementary material Table S1. Textual information on relevant policy documents issued by the Chinese government from 2016 to 2022).

**Table 1.** Summary of relevant policy texts on online ride-hailing policies issued by the central government (excerpt).

| Number | Title | Department | Test Number | Date |
|--------|-------|-----------|-------------|------|
| 1 | Guidance of the General Office of the State Council on Deepening Reform and Promoting the healthy Development of the taxi Industry | Ministry of Transport | No. 58 Document in 2016 of the General Office of the State Council of the People's Republic of China | 26 July 2016 |
| 2 | Notice of the General Office of the Ministry of Transport, the General Office of the Ministry of Industry and Information Technology, and the General Office of the Ministry of Public Security on the working process of online car-hailing operators applying for online service ability identification | Ministry of Transport, Ministry of Industry and Information Technology, and Ministry of Public Security | No. 143 Document in 2016 of the General Office of the Ministry of Transport | 3 November 2016 |
| 3 | Notice of the General Office of the Ministry of Transport on online car-hailing access and exit | Ministry of Transport | No. 144 Document in 2016 of the General Office of the Ministry of Transport | 7 November 2016 |

**Table 1.** *Cont.*

| Number | Title | Department | Test Number | Date |
|--------|-------|-----------|-------------|------|
| 40 | Notice of the State Council on issuing the Overall Work Plan for Epidemic Prevention and Control and Transportation Service Guarantee during the Spring Festival Transportation in 2022 | Joint prevention and control mechanism for the novel coronavirus pneumonia outbreak | No. 3 Document in 2022 of the General Office of the Ministry of Transport | 6 January 2022 |
| 41 | Notice of the General Office of the Ministry of Transport, the General Office of the Ministry of Industry and Information Technology, and the General Office of the Ministry of Public Security on strengthening the joint supervision of the whole chain before, during and afterward of the online car-hailing industry | Ministry of Transport, Ministry of Industry and Information Technology, Ministry of Public Security, Ministry of Human Resources and Social Security, People's Bank of China, State Administration of Taxation, State Administration for Market Regulation, Cyberspace Administration of China | No. 6 Document in 2022 of the General Office of the Ministry of Transport | 7 February 2022 |
| 42 | Notice of the China Banking and Insurance Regulatory Commission and the People's Bank of China on Strengthening Financial Services for New Citizens | The China Banking and Insurance Regulatory Commission, the People's Bank of China | No. 4 Document in 2022 of the China Banking and Insurance Regulatory Commission | 4 March 2022 |
| 43 | The General Office of the Ministry of Transport and the Ministry of Transport issued the notice of five practical work plans closer to people's livelihood, including aging transportation services in 2022 | Ministry of Transport | No. 495 Document in 2022 of the General Office of the Ministry of Transport | 29 March 2022 |

At present, most research on policy system construction analyzes policy texts from a two-dimensional perspective, and examines the overall appearance of policy system construction in specific fields through the binary matching of policy text itself, policy target, implementation process, historical evolution, and other dimensions. On this basis, the current study selects three dimensions, namely, policy instruments, policy development, and the TOE scenario, to construct a policy analysis framework for online ride-hailing in China. We analyze the current effective online ride-hailing policy under the three-

dimensional research framework and explore the style and characteristics of this policy system based on the three-dimensional framework.

*3.1. X Dimension: Basic Policy Instruments*

Policy instruments are a series of policy sets with specific goals to achieve policy goals, and are classified in many ways. For example, based on the degree of government coercion, Lowie, Dahl, and Lindblom, classified policy instruments into regulatory and non-regulatory instruments, while Michael Howlett and M. Ramesh divided them into voluntary, mandatory, and mixed categories [27]. Lorraine M. McDonnell and Richard F. Elmore classified policy instruments into imperative, incentive, capacity building, and system change tools, according to the purpose of their use [28]. Chinese scholars Gu Jianguang and Wu Minghua divided policy instruments into three categories: control, incentive, and information transmission policy instruments, based on the application method of the policy instruments [29]. To examine the characteristics of online ride-hailing policies and the adaptation of different classification methods, this paper selects the policy instruments classification method proposed by Gu Jianguang and Wu Minghua, and divides the policy instruments into three categories: control, incentive, and information transmission.

We believe that to make a more detailed classification of the online ride-hailing policies, it is necessary to have a clear understanding of the object of these policies. From the perspective of policy development, online ride-hailing policies have specific objectives at different stages of development. The online ride-hailing policy mainly includes three categories: platform companies, drivers, and the public. Further, the policy objectives set by the government based on market demand are also different in different periods. In the process of public policy formulation in China, the values and the central-local government system tend to have a bearing on policy goals. The values could systematically and interactively have a notable impact on the whole process of the input of public demand information, the cognition of public demand information, the rotation of public demand information, and the output of public policy [30]. Under the guidance of values such as "putting the people first", "a good life for the people", and "stabilize employment and ensure people's well-being", the Chinese central government is likely to formulate policies with different characteristics according to the needs of social development, such as ensuring social security, maintaining market order, and promoting employment. Meanwhile, the implementation of public policy in China takes place in a unique political ecology, in which the central government always acts as a promoter. Compared with the policies formulated by local governments, the policies formulated by the central government are more guiding and holistic. Therefore, the change of policies formulated by the central government directly reflects the governance focus and policy goals in a certain period.

In the early stage of the development of online ride-hailing, the government might pay more attention to the supervision and management of platform companies to create a more standardized market. The government might ignore the various problems that drivers and the public experience in the ride-hailing market. However, with the continuous expansion of the online ride-hailing industry, the natural drawbacks of the market made it unsuitable for governments to focus on the online ride-hailing platform only, for it could no longer meet the needs of development. Therefore, the government would inevitably pay more attention to the response of online ride-hailing drivers and public demands to achieve certain goals. It could be found that China's online ride-hailing policy has the characteristics of multiple objects and dynamic changes in goals with a further analysis at different stages. That is to say, we can properly combine these two structural characteristics and specifically classify the content of China's online car-hailing policy according to the way of policy is being used.

Based on the comprehensive characteristics of the online ride-hailing policy structure, and the objects of policy function under market conditions, we found that the policy instrument method proposed by Gu Jianguang and Wu Minghua is highly compatible with the content and structure of China's online ride-hailing policy. These policy instruments

divide the policy instruments into three categories: control, incentive, and information transmission, which helps us to make an in-depth analysis of the development process of online ride-hailing policies more logically. To provide clearer proof of the applicability of these policy instruments, it is necessary to further illustrate the specific implications of each class. The purpose of regulatory policy instruments, which has the characteristics of mandatory and responsiveness, is to regulate the behavior of the social actors. The purpose of incentive-based policy instruments is to exert influence on people's behavior through positive or negative incentives, so that they can achieve the goal expected by the government. Economic subsidies, social security, and other policies, belong to this category of policy instruments. As far as new energy subsidies are concerned, they are not only good for the environment but also for business performance. This is because active green process innovation by companies improves their short and long-term corporate financial performance [31,32]. Information transfer policy instruments are an important link between policy executors and target groups. A policy instrument can only bring about change with respect to the target actors when policy information reaches the target group correctly and effectively, and is accepted by it. These policy instruments aim to create effective information communication and sharing channels for the development of the online ride-hailing industry, and promote the standardized development of the industry. Therefore, their role targets include platform companies, drivers, and the public. The government strengthens the practical effectiveness of the policy by clarifying relevant concepts and laws, standardizing the rights and obligations of all subjects, building a dispute resolution mechanism, and strengthening public publicity.

　　　Based on the above analysis, this paper divides the online ride-hailing policy into three major categories and 16 minor categories, according to three different policy instruments (control policy instruments, incentive policy instruments, and information transfer policy instruments), as shown in Table 2.

**Table 2.** Name and meaning of policy instruments.

| Type | Name | Definition |
|---|---|---|
| Control instruments | Direction setting | Promote the overall healthy development of the online ride-hailing industry |
| | Standard formulation | Industry access, elimination mechanism |
| | Industry regulation | In addition to finance and information, to supervise and control various matters in the operation process of online ride-hailing |
| | Financial regulation | Price, deposit, drawing specification |
| | Information regulation | Collect users' information |
| | Emergency control | Special Emergency management during the COVID-19 |
| Incentive instruments | Industry development | Emphasize the overall innovation of platform economy and "Internet+" industry |
| | Market shaping | Build new forms of integrated development of cruise vehicles and online ride-hailing services |
| | Social security | Employment, social insurance, commercial insurance, and suitable for the elderly function, etc. |
| | Tax rent discount | State financial support for the resumption of work and production during the COVID-19 |
| | Green economy | Subsidies for new energy vehicles |
| Information Transfer instruments | Clear nature | Standardize the concept, service scope and applicable laws of online ride-hailing services |
| | Scope of rights and responsibilities | Clarify the rights and obligations of the platform, drivers, and passengers |
| | Information sharing | Establish the information exchange mechanism of online ride-hailing supervision, and play the role of social supervision based on information disclosure |
| | Dispute settlement | Complaint alarm and rapid response mechanism |
| | Public propaganda | Strengthen news publicity and public opinion guidance |

*3.2. Y Dimension: Policy Development*

　　　To vertically analyze the use of online ride-hailing policy instruments, and explore the development and change of the style characteristics of online ride-hailing policies in China, this study selected the policy development process as the Y analysis dimension. With the

help of the website called PUKLAW (https://www.pkulaw.com/, All accessed on 9 August 2022), the policy texts issued by the central government increased year-on-year since 2016, such that the number of policies even exceeded 40 in 2018, 2020, and 2021. Through further analysis of the content of the policy text, it was found that the release of the "Guiding Opinions of the General Office of the State Council on Deepening Reform and Promoting the Healthy Development of the Taxi Industry" in 2016 marked the point at which the central government began to recognize the compliant development of online ride-hailing. After a series of safety accidents in 2018, the central government issued an "Emergency Notice on Further Strengthening the Safety Management of Online Taxi Reservation and Private Passenger Cars", emphasizing a comprehensive review of drivers with existing online ride-hailing and private minibus sharing services. Under the dual pressure of an increasingly complex market environment and a security crisis, the government has paid more attention to the standardized development of online car-hailing. Since the outbreak of COVID-19 in 2020, the ride-hailing industry has created a large number of jobs for society; therefore, the government pays more attention to this industry. Based on the above analysis, we can roughly divide policy development since 2016 into three stages: the order construction period in 2016–2018; the prudent regulation period in 2018–2020; and the standardized development period from 2020.

### 3.3. Z-Dimension: The TOE Scenario Feature

The development of the online ride-hailing industry and the formulation of related policies are comprehensively influenced by information technology, market tolerance, and the development environment of online ride-hailing. Therefore, this study took the TOE scenario feature as the Z analysis dimension and depicted the impact of the current online ride-hailing policy within the context of different scenario-based dimensions. The TOE framework was proposed by Louis G. Tornatizky and Mitchell Fleischer in 1990, and was first applied to analyze the influencing factors for the adoption of new technologies [33]. Later, this framework was widely used in research on the adoption and implementation process of new strategies, new policies, and new business models, with strong systematization, flexibility, and operability. The model focuses on the action mechanism of the multilevel technology application scenario, with respect to the technical realization effect. Examples include the specific application scenario of technology, the degree of organizational demand for technology, and the application degree of technology and the organizational framework [34]. Technical conditions refer to the characteristics of the technology itself and its relationship with the organization. It mainly depends on whether the technology matches the structural characteristics of the organization, whether it is coordinated with the application ability of the organization, whether it is compatible with the application ability of the organization, and whether it can bring benefits to the organization, or potential benefits [35]. The influence of organizational conditions on the application of technology mainly includes the organizational scale, business scope, formal or informal institutional arrangements, communication mechanisms, and idle resources of reserve savings [36]. Environmental conditions include the market structure of the organization and the external control policies of governments [37]. From the perspective of technical conditions, in order to improve the positive role of information technology in the development of the car-hailing industry, the policies issued can be divided into two major technical scenarios: standardized technology application; and technology empowerment development. The organizational situation can be understood as the central government's emphasis on the car-hailing industry and its attitude towards the development of the industry. Therefore, organizational conditions can be divided into two scenarios: active promotion; and strict control. Environmental conditions can be analyzed from both macro and micro dimensions. First, the macro environment includes the influence of the government's overall goal of promoting economic development and maintaining social stability, which can be understood as the background for the introduction of the car-hailing policy. Second, the micro environment refers to the citizens' needs for travel, employment, and social security, as well as the practical

needs of enterprise development and innovation for supporting policies. Overall, the TOE framework provides six different scenarios based on the three first-level conditions, which helps us to further understand policy style regarding online ride-hailing in China from the perspective of scenario feature, which is composed of technology, organization, and environment (as shown in Figure 1).

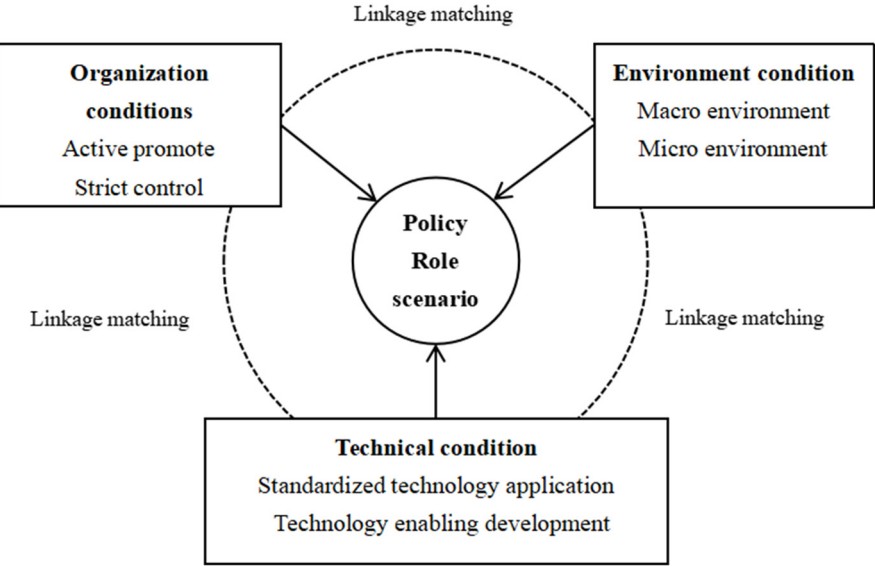

**Figure 1.** TOE scenario feature dimension.

### 3.4. Three-Dimensional Analytic Framework of the Online Ride-Hailing Policy Text

Starting from basic policy instruments, this study integrated the development process of online ride-hailing policy and the TOE scenario analysis to construct the X-Y-Z three-dimensional analysis framework (as shown in Figure 2), and then conducted a quantitative and multidimensional cross-analysis of the policy text.

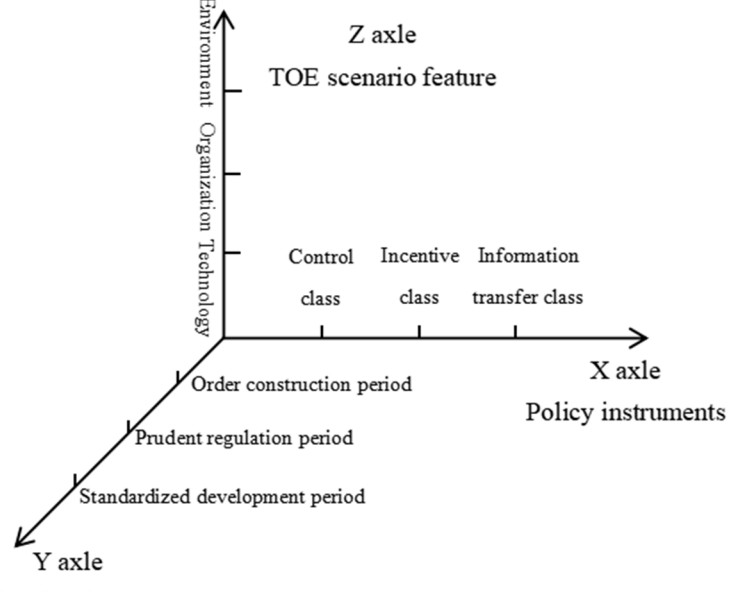

**Figure 2.** X-Y-Z three-dimensional analysis framework.

In addition, to ensure mutual exclusivity between classes to safeguard reliability, we analyzed 43 policy text items that had been screened. Conforming to the three-dimensional pol-

icy analytic framework, we then coded the items one by one. General standard documents, such as the "Notice of the General Office of the Ministry of Transport on the Access and Exit of Online Taxi Reservation", were not divided into secondary codes. Finally, we obtained 109 text cells (For coding details see Supplementary material Table S2. The process of coding policy texts) and used "Excel 2016_64bit" (Microsoft Corporation, Redmond, WA, USA) to count the encoding results (as shown in Table 3).

**Table 3.** Coding of policy text content analysis unit (excerpt).

| No. | Content Analysis Unit of the Policy Text | Code |
|---|---|---|
| 1 | (2) Adhere to reform and innovation. Seize a favorable opportunity to implement the "Internet+" action, adhere to a problem-oriented approach, promote the transformation and upgrading of cruise taxis, standardize the operation of online taxi booking, and promote the integrated development of both forms of business. | 1-1 |
| | (2) Adhere to the overall consideration. Coordinate public transport and taxis, innovation and development, safety and stability, the development of new and old business forms, the interests of passengers, drivers, and enterprises, and promote reform step-by-step and prudently. | 1-2 |
| | (3) Posit taxi service scientifically. Taxi services mainly include cruises, online booking, and other ways. | 1-3 |
| | (7) Promote industry transformation and upgrading. Encourage cruise car operators and online ride-hailing operators to implement corporate operations in accordance with the modern enterprise system through mergers, reorganization, absorption, and investment, so as to realize the integrated development of new and old business forms. Encourage cruise car companies to transform themselves to provide online ride-hailing services. | 1-4 |
| | (8) Standardize the development of online ride-hailing services. | 1-5 |
| | (9) Standardize the business behavior of online ride-hailing services. | 1-6 |
| | (10) Standardize the sharing of private passenger cars. The people's government of the city shall encourage and standardize its development, formulate corresponding regulations, and clarify the rights and obligations of the driver, passenger, and information platform company. | 1-7 |
| 2 | According to the Online Ride-hailing Service Management Interim Measures (Ministry of Industry and Information Technology, Ministry of Commerce, Ministry of Industry and commerce administration order 60 in 2016), to optimize services, standardize operations, ensure convenient online ride-hailing operators, the work process regarding online service capacity identification for online ride-hailing operations is hereby notified as follows. | 2-1 |
| 3 | According to the Road Traffic Safety Law of the People's Republic of China and Online Ride-hailing Service Management Interim Measures and other laws and regulations, to facilitate services, simplify procedures, and standardize management, after consulting with the Traffic Administration Bureau of the Ministry of Public Security, the access and exit of the relevant work process of online ride-hailing is notified as follows. | 3-1 |
| 41 | 1. Further improve the working mechanism of joint supervision. | 41-1 |
| | 2. Strengthen the joint supervision of the whole chain before, during and after the event. | 41-2 |
| | 3. The joint supervision of the whole chain before, during and after the event. | 41-3 |
| | 4. Joint supervision and disposal process of the whole chain before, during and after the event. | 41-4 |
| | 5. Other work requirements. (1) Strengthen the coordination between all parties. (2) Strengthen emergency response. (3) Innovate ways of supervision. | 41-5 |
| 42 | (7) Improve the level of insurance for new citizens to start businesses and find jobs. Focus on new citizens, such as construction workers, express delivery riders and prominent professional risks, and ride-hailing drivers. | 42-1 |
| 43 | 1. Continue to improve the service level of cruise taxi calling, and improve the "one-click ride-hailing" and telephone calling function and service response of online ride-hailing. | 43-1 |
| | 2. Urge major ride-hailing platform companies and Internet road freight platform companies to disclose the pricing rules to the public, reasonably set the proportion or membership fee and publish them; display the percentage ratio or information service fee on the driver side. | 43-2 |

## 4. Policy Style Analysis of Online Ride-Hailing Policy Based on Three-Dimensional Analytic Framework

The formation of their own policy networks is realized by the multiple governance subject elements involved in the online car-hailing policy. At the core of the network structure, the government controls resource allocation, in which the regulatory authorities have some resources and the right to speak. Furthermore, the online car-hailing policy network and market relations have been influenced by the continuous development of digitalization and the external environment. In this case, the vertical and horizontal vulnerabilities of the system are highlighted, and the policy system with strong autonomy may make the system structure gradually develop towards the direction of unstable situation. This causes the main elements of the policy to undergo adaptive interaction and order breakthroughs in the allocation of resources, as well as the acquisition and exercise of power. The original closed policy network structure should be overcome, the governance radius of the policy network

should be expanded, and the three-dimensional policy instruments and TOE framework evaluation should be adopted to truly make the public become the "destination" of the policy network. As a result, the theoretical framework of "institution-function" exists. The "resilience" of the policy network is kept by putting people first. This can help the formation of a localized online car-hailing policy network system with strong integration, a high degree of system openness, and the interdependence of internal networks. In this section, we will analyze the online ride-hailing policy style in China from an analytical framework constructed from three dimensions: policy instruments, policy developments, and the TOE framework.

### 4.1. Analysis of Basic Policy Instruments Dimension

According to the classification of basic policy instruments, the distribution results of basic policy instruments of online ride-hailing are shown in Table 4 and Figure 3. Among the 109 policy texts, control class policy instruments accounted for the highest proportion, i.e., 55.5%. These were followed by incentive class policy instruments accounting for 34.86%, while information transfer class policy instruments accounted for the lowest proportion, at 10.09%. Among control class policy instruments, industry regulation was identified as the main tool, accounting for 51.67%. Meanwhile, direction setting tools also represented a large proportion, accounting for 26.67%. Among the incentive class policy instruments, the number of economic and social policies was the same, among which social security policies accounted for 42.11%, and other policies such as industry development, market shaping, tax rent concessions, and the green economy, fall under the category of economic policies. The number of different types of tools in the information transfer class policy instruments was similar, among which there were many information sharing tools, accounting for 45.45%.

**Table 4.** Distribution table of basic policy instruments in the online ride-hailing policy text.

| Type | Name | Policy Number | Number | Proportion | Total Proportion |
|------|------|---------------|--------|------------|------------------|
| Control class | Direction setting | 1-5; 1-6; 10-3; 10-12; 36-7; 36-8; 36-11; 36-12; 39-1; 39-2; 39-3; 39-4; 39-7; 39-9; 39-16; 39-17 | 16 | 26.67% | 55.05% |
| | Standard formulation | 2-1; 3-1 | 2 | 3.33% | |
| | Industry regulation | 5-1; 6-1; 7-1; 7-2; 8-1; 9-1; 9-2; 9-3; 9-5; 10-4; 10-5; 10-6; 10-7; 10-13; 23-1; 26-1; 28-1; 31-1; 33-1; 34-1; 34-2; 34-3; 34-5; 37-1; 38-1; 40-1; 41-1; 41-2; 41-3; 41-4; 41-5 | 31 | 51.67% | |
| | Financial regulation | 10-1; 11-1; 14-1; 29-1; 36-1; 39-5 | 6 | 10.00% | |
| | Information regulation | 13-1; 27-1; 39-6 | 3 | 5.00% | |
| | Emergency control | 17-1; 18-1 | 2 | 3.33% | |
| Incentive class | Industry development | 1-1; 1-2; 15-1; 24-1; 29-2; 30-1; 39-8; 39-11; 39-12; 39-13; 39-14; 39-15; 39-18 | 13 | 34.21% | 34.86% |
| | Market shaping | 1-4; 10-2; 10-8; 12-1; 16-1; 43-1 | 6 | 15.79% | |
| | Social security | 19-1; 22-1; 25-1; 32-1; 32-2; 32-3; 34-4; 36-2; 36-3; 36-4; 36-5; 36-6; 36-10; 38-2; 39-10; 42-1 | 16 | 42.11% | |
| | Tax rent discount | 20-1 | 1 | 2.63% | |
| | Green economy | 21-1; 35-1 | 2 | 5.26% | |
| Information transfer class | Clear nature | 1-3 | 1 | 9.09% | 10.09% |
| | Scope of rights and responsibilities | 1-7 | 1 | 9.09% | |
| | Information sharing | 4-1; 10-9; 10-10; 14-2; 43-2 | 5 | 45.45% | |
| | Dispute settlement | 9-4; 36-9 | 2 | 18.18% | |

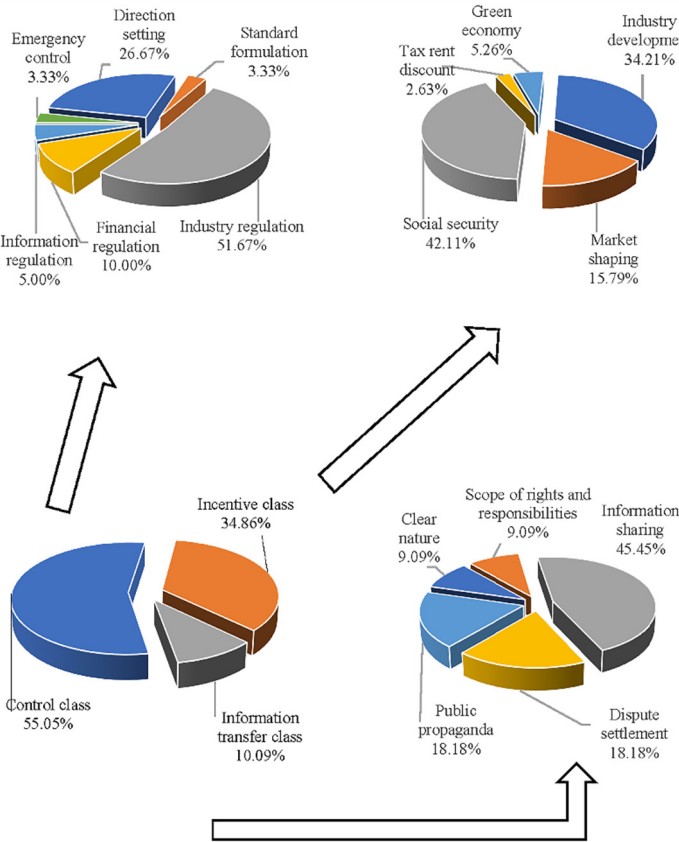

**Figure 3.** Schematic diagram of the percentage of basic policy instruments used in the online ride-hailing policy text.

Thus, we can identify three features of the X-dimensional distribution of online ride-hailing policies. First, there are excessive regulatory policy instruments. At present, the policies issued by the central government mainly focus on supervision and management policies, and relatively few policies to promote innovation, social welfare, and informatization. The online car-hailing industry is an emerging industry, and problems are prone to occur in the process of coordinated development with the traditional taxi industry. Therefore, the current policy is mainly a regulatory policy, and the main purpose is to regulate the development of new business formats. Especially after the proclamation of the "Guiding Opinions of The General Office of the State Council on Deepening the Reform and Promoting the Healthy Development of the Taxi Industry", the central government has clarified the basic direction of the overall development of the traditional taxi industry and online ride-hailing services; that is, to "coordinate the development of cruise taxi and online ride-hailing, implement dislocation development and differentiated operation, and provide quality and diversified transportation services for the public" [38]. Under such a policy background, in order to promote the integrated development of new and traditional business forms, it is necessary to improve the market supervision mechanism, determine industry access standards, and standardize the behaviors of online ride-hailing platforms, drivers, passengers, and labor companies. At the same time, due to the particularity of the online car-hailing business model, there are still many institutional blind spots in its operation process. Therefore, the government hopes to strengthen the control of the online car-hailing industry, aiming to promote the orderly development of new formats.

Second, the information transfer class policy instruments are defective. In terms of quantity, compared with the two policy instruments of the control category and the incentive category, the information transmission policy instruments are very lacking. On the one hand, practitioners in the ride-hailing industry can fundamentally promote the standardized development of the ride-hailing industry only after clearly understanding

the attitude of the central government towards the development of ride-hailing, and the relevant systems and norms of the industry. Otherwise, poor information transmission will not only be detrimental to the effective implementation of regulatory policies, but also have a negative impact on the overall orderly development of the industry. On the other hand, a good information communication mechanism is the basis for promoting the healthy development of the industry. As a passenger service, the basic principle of online car-hailing is to ensure the safety of passengers and drivers, and to pay attention to the travel experience of passengers. Therefore, a sound complaint response mechanism can be an important guarantee for promoting the healthy development of the car-hailing industry.

Third, the number of incentive policy instruments is appropriate and abundant. Compared with the control category and information transfer policy instruments, the number of incentive policies is moderate. The online ride-hailing industry can create a large number of jobs for society, and generate employment for residents while promoting economic development. At the same time, the development of online car-hailing can drive the development of the green economy, increase the load rate of empty vehicles, and optimize the supply and demand allocation of urban transportation resources. For example, the Notice of the Ministry of Finance, the Ministry of Industry and Information Technology, the Ministry of Science and Technology, the Development and Reform Commission on Improving the Financial Subsidy Policy for the Promotion and Application of New Energy Vehicles, and the Ministry of Transport's issuance of the "14th Five-Year Development Plan for Green Transport", have incorporated online ride-hailing into the specific goals of the development of new energy vehicles.

*4.2. Analysis of the Policy Development Dimension*

To further analyze ride-hailing policy styles at different stages of development, it is helpful to incorporate the time factor into the analysis of different policy instruments. Based on the previous analysis, we took the promulgation of two documents entitled the "Guiding Opinions of The General Office of the State Council on Deepening the Reform and Promoting the Healthy Development of the Taxi Industry" and "Urgent Notice on Further Strengthening the Safety Management of Online Taxi Reservation and Private Passenger Cars", and the COVID-19 outbreak as the time nodes, to classify policy development from 2016 to 2022 into three stages. Figure 4 illustrates the different combinations of policy instrument types and their evolution at different stages of development. From 2016 to 2018, the number of policy instruments was relatively small for each of the three categories, and the legal status of online ride-hailing had just been granted, which means that it was in the initial stage of development. At this stage, the government mainly promoted the development of the car-hailing industry by giving full play to the autonomy and power of the market. Instead, overall plans were communicated to outline the development direction of online ride-hailing. In 2018–2020, the number of control policy instruments increased significantly. Because for one thing, increasingly more negative passenger experiences have occurred in the online ride-hailing industry, and the personal safety of passengers has attracted the attention of the government and the public. On the other hand, after several years of free development, the online car-hailing industry has formed a solid market foundation and a monopoly position, which requires the government to introduce regulatory measures. From 2020 to 2022, while the number of regulatory policy instruments continued to increase, the number of incentive-based policies also showed an obvious growth trend and was six times greater than that of the previous stage. Against the backdrop of the seemingly unending COVID-19 epidemic, the global economy is continuing to experience tremendous downward pressure, and high unemployment has generated instability in several areas of society. Therefore, to address the issue of unemployment and promote the economy, the government has encouraged the online ride-hailing and food delivery industries to create more jobs, appealing to platform enterprises to protect the legitimate rights of workers. The "Opinions of the Ministry of Transport", the Publicity Department of the CPC Central Committee, and the Cyberspace Administration of the

CPC Central Committee on Strengthening the Protection of the Rights and Interests of Employees in New Forms of Transport, set forth detailed provisions on the protection system for the rights and interests of employees, as well as the employment environment. At the same time, based on the growing demand of the digital economy and green economy, the government has also increased its economic support for the car-hailing industry.

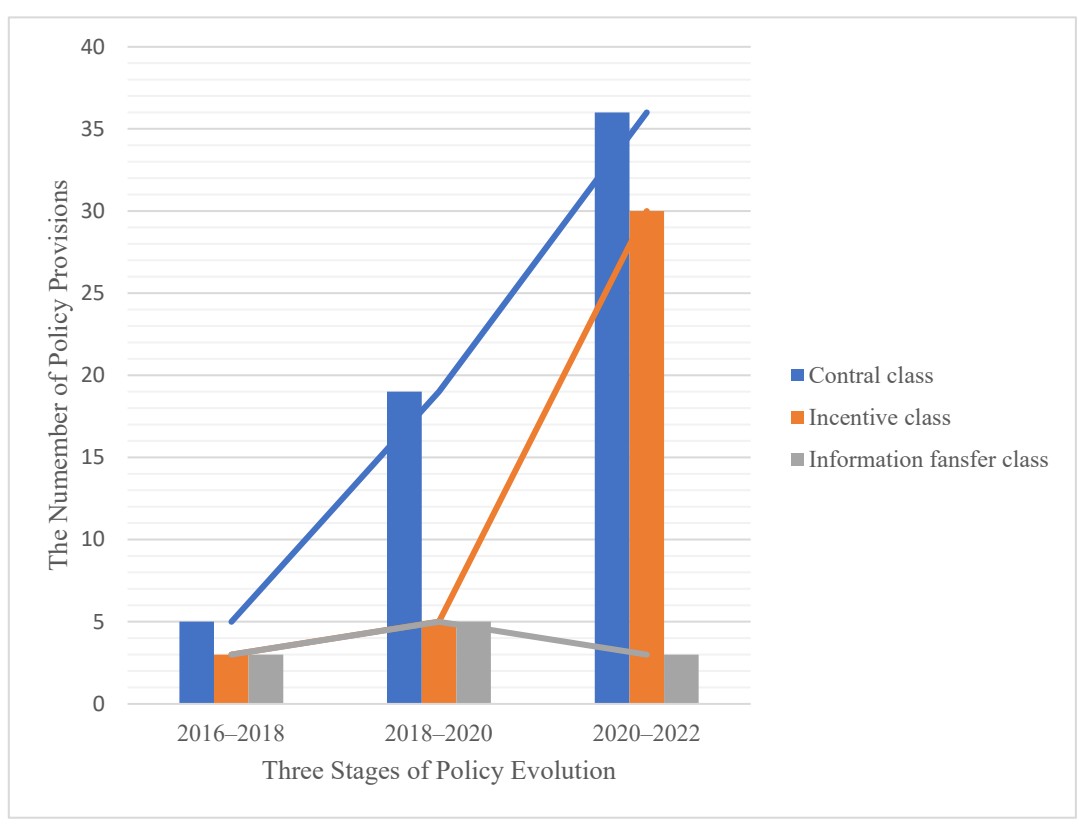

**Figure 4.** Differences in the number of policy instrument types at different development stages.

*4.3. Analysis of the TOE Scenario Feature Dimension*

Analyzing the background of policy implementation can help us better understand a given policy and clarify the factors that influence the issuance of policies. First of all, from a technical perspective, the development of the online ride-hailing industry has necessitated the collection of vast amounts of private customer and passenger information, the illegal collection and use of which has become a serious issue. Therefore, in order to protect citizens' privacy and reduce the risk of information leakage, the state has clearly stipulated the scope and authority of online car-hailing platforms to collect passengers' personal data. In this context, the central government issued the "Practice Guide for Network Security—Necessary Information Specification for Basic Business Functions of Mobile Internet Application", "Provisions on the Scope of necessary Personal Information for Common Type of Mobile Internet Applications", and other documents. Detailed regulations have been formulated to address data collection and use issues among network operators, including those in the online ride-hailing industry. At the same time, technological progress can also promote the development of the car-hailing industry. Therefore, relevant policies provide sufficient impetus for the development of the industry by encouraging the innovation of online ride-hailing platforms. Moreover, an online ride-hailing information supervision and sharing platform at the central level should play a more important role. An organization's influence on policy can also be seen as a time-varying process, influenced by both internal and external factors. In the early days of the industry's development, the central government and all sectors of society held different views on the legality and ownership of the car-hailing industry, which explains why the central government rarely issued relevant

policy documents for the development of car-hailing. From 2012 to 2016, there was no clear guidance on the development of the car-hailing industry. After the online ride-hailing industry reached a certain market scale, problems began to emerge, particularly in relation to its monopolist position and non-standard operation, which adversely affected the overall development of the taxi industry. At this time, the government's attitude towards online ride-hailing shifted to building order and promoting the standardized development of the industry, and promulgating a series of policies to supervise the car-hailing industry. After a period of correction in this industry, the development of online ride-hailing became increasingly more standardized, and the government has gradually paid attention to the social security concerns of online ride-hailing employees. At the same time, COVID-19 has affected the economies of countries around the world, and it is a challenge for governments to address the issue of massive worker unemployment [39]. As a result, a series of policies have been promulgated to promote employment and good levels of social welfare for online ride-hailing employees, while considering realistic demands for economic development and greater employment opportunities in the post-epidemic era. Environmental conditions can be understood as indirect variables that affect the introduction of policies by affecting organizational attitudes. The requirements of economic development and social stability at the macro level, as well as travel, employment, and social security at the micro level, also affect the attitude of government organizations towards the online ride-hailing industry, which in turn impacts government decision-making. For instance, as human energy consumption increases, developing and developed countries alike are facing increasing ecological fragility and scarcity of natural resources [40]. With environmental issues becoming a growing concern for governments, the Chinese government has also implemented a series of policies to promote the green development of the online ride-hailing industry [41].

*4.4. Overall Style Analysis of the Central Online Ride-Hailing Policy*

Policy mix is "a long-lasting arrangement of policies that have accumulated over time [42]", and the policy mix consists of the policy logic of the specific policy participants [43]. Therefore, in response to a change in policy logic, policy style will also change. The logic of policy formulation is the comprehensive result of technology, organizations, the environment, the development stage, and other factors in the above analysis, which are characterized by dynamic change. At the same time, the attitude of high-level rulers to China's political environment is also an important factor affecting the policy style. The review of online ride-hailing policies in 2016–2022 and the three-dimensional analytic framework revealed that, in regard to online ride-hailing, the overall characteristics of a policy-making style largely adopted control policy instruments, which were supplemented by incentive policy instruments. China's socialist economic system emphasizes both the government and the market. As such, the Chinese government often regards supervision and management as the core tasks of the government based on a consideration of legitimacy, order, and conservatism, in the process of policy formulation. Overall, China's car-hailing policy is in line with the government's long-standing policy-making style for emerging industries and has played an active regulatory role. However, due to the restraining effect of strict supervision on industry innovation, there is still room for optimization of this policy mix.

**5. Discussion**

Based on our overall analysis of China's online ride-hailing policy, the following conclusions can be drawn: first, the current regulatory policy instruments account for the largest proportion of the online ride-hailing policy mix, and information transmission policy instruments are very lacking; second, the policy development process of online ride-hailing can be roughly divided into three stages. Although the content of the policy portfolio at each stage is affected by multi-dimensional factors such as technology, organization, and environment, the overall policy style features have the characteristics of regulatory policy instruments; finally, the government's economic and social incentive policies for the

development of the car-hailing industry have gradually increased, which is conducive to promoting the integrated development of new business formats and traditional industries. At the same time, car-hailing has certain advantages in the green sharing economy.

Further analysis of the above quantitative results allows us to conclude that the current Chinese central-level policy on online ride-hailing still faces a series of problems and there is room for further development. First, the single type of policy instrument is not conducive to the overall coordinated development of online ride-hailing. This diversity of participants requires different types of policy instruments and policy provisions with different purposes to regulate their behavior and determine their rights and responsibilities. If the government only focuses on using regulatory policy instruments to manage the online ride-hailing industry, it will tend to neglect the protection of workers' and consumers' rights, which is not conducive to the healthy and sustainable development of the industry. Second, while we agree that there is competition between online ride-hailing and traditional taxis, this does not mean that the two are incompatible. On the contrary, the emerging and traditional taxi industries should have similar development goals and take advantage of integrated development. In reality, however, the traditional taxi development model of China still maintains the offline soliciting and stop-on-wheels mode of operation, which lacks competitiveness compared to online ride-hailing. Although the current Chinese central government has documented the need to support the integration of the two industries and promote technological innovation in the traditional taxi industry, the only relevant policies could be categorized as policy instruments at the level of directional guidance and the lack of substantive policies to support the integration of the two industries. Obviously, the lack of incentive policy instruments, is an important reason why the integration of these two industries has been difficult to achieve for a long time. Thirdly, the style of policy at the central level in China is different at different stages of development of the online ride-hailing industry, with the aim of achieving different development goals at different stages. We can then further argue that different styles of policy objectives can also be developed in different regions to suit the needs of local economic development, as well as diverse passenger travel needs. Due to the imperfection of the current online car-hailing joint policy formulation system, the high degree of policy fragmentation, and the lack of specific measures, etc., this paper puts forward the following policy and market optimization direction:

(1) There is a need to improve the policy system of online ride-hailing by reasonably allocating the proportion of the three policy instruments corresponding to control, incentive, and information transmission. In addition, it is necessary to pay more attention to the richness of different types of internal policies, and to promote the healthy development of the car-hailing industry in multiple dimensions. Cross-departmental cooperation is the organizational premise for diversified policies. Only by making concerted efforts in the fields of transportation, economy, social security, safety, and culture, can we promote the improvement of the multi-level car-hailing policy system. The richness of policies is also reflected in the replacement of flexible policies according to local conditions, and it is necessary to provide different incentives for the development of the car-hailing industry in different regions. For example, in some cities with poor public transportation, the development of the ride-hailing industry can be encouraged by jointly developing services between the government and online ride-hailing platforms. In areas with a well-established urban public transport system and a high number of private cars, it is necessary to appropriately limit the development scale of online car-hailing to provide policy support for alleviating traffic congestion and protecting the environment [44]. In addition, special attention should be paid to improving information transmission policy instruments, making the online car-hailing market more open and transparent, and promoting the standardized development of the industry. For one thing, it is necessary to build a clear rights and responsibilities relationship between the platform, drivers, and passengers, so as to resolve all forms of disputes promptly and effectively. For another thing, the

government should make efforts to enhance policy interpretation. It will contribute to the implementation of policies and the formation of healthy government-market relations. Future research can evaluate the implementation effects of car-hailing policies in different regions by means of configuration analysis, exploring the influencing factors of the different effects of the same policy in different regions, and explaining the reasons for the construction of car-hailing.

(2) For the overall development goal of the integration of traditional cruise taxis and online ride-hailing, the specific operation mode should be detailed. In order to effectively cope with the dilemma of integrating development between online ride-hailing and cruise taxis, first of all, it is necessary to correctly locate the positioning of the car-hailing industry. If the ride-hailing industry is seen as a complement to public transport, policies should be introduced to integrate it into the public transport system [44]. If car-hailing is only defined as an alternative means of transportation, the competition between car-hailing, buses, subways, and traditional cruise taxis, should be carefully considered. Therefore, it is wiser to build a multilevel and differentiated transportation system. Policymakers should also acknowledge the huge impact of online ride-hailing on the traditional cruise taxi industry. This means they must fundamentally address the contradiction in the market share of these two types of taxis. Furthermore, governments can also reduce the loss of profit for traditional industries; for instance, technology empowerment and market adjustment are effective methods. Finally, we can aim to change the business model of cruise taxis and narrow the differences between these two types of industries in information acquisition and operation mode. In the future, we should focus on the research on the integrated development path of cruise taxi service and online car-hailing. Moreover, it is important to examine how the dilemma of integrated development, such as information sharing barriers, inconsistent industry standards, and inevitable interest competition, can be resolved by considering multiple dimensions.

(3) The construction of the car-hailing policy system should focus on the needs of industry development and social and economic development. Governments should improve the relevant policies promptly according to actual needs, so that the policy style is in line with the development goals. To be sure, relevant policies have played an effective role in regulating the chaotic development of the online car-hailing market. After that, some policies supporting the development of ride-hailing have promoted the development of the industry. An adaptable policy style is critical to ensuring that the policy system has both economic and social utility. Therefore, in the subsequent process of policy formulation, it is also necessary to conduct a comprehensive investigation of various multidimensional factors, such as technology, organization, and the environment. Multidimensional analysis can help to identify specific policy objectives, thereby forming a stable and adaptable policy style. In the future, neural network analysis and machine learning methods can be used to predict online ride-hailing policies, and provide intellectual support for policymakers.

(4) With the advancement of industrialization and urbanization in China, contradiction between taxi companies and online ride-hailing is increasingly prominent; conflicting events between online ride-hailing security incidents and online ride-hailing consumes' interests, frequently occurs. The multiple and complex online ride-hailing problems and the traditional governance model of the government make the social stability, government credibility, and public satisfaction of online ride-hailing governance in China suffer continuous doubts and impacts. This situation even evolves into major mass incidents, which have led to the loss of a large amount of government political benefit and the formation of a new local development crisis. The governments use various institutional political resources, such as institutional systems, policies, and regulations, to regulate and constrain the relationships and behaviors among various governance entities. These laws and policies belong to the institutional system of online ride-hailing governance, and the institutional government benefit of

the government to achieve the online ride-hailing market sustainability. The loss of political gains for the government will lead to reduced economic gains. The decline in government authority and legitimacy has an impact on social stability, business environment, and policy implementation, which hinders local economic development. There is a law of reciprocity between government political gains and economic gains in online ride-hailing governance. The political impact on various interest groups caused by the waste of government political benefit, and the excessive consumption of government political benefit, will endanger the legitimacy of the government and political security and stability. Once that is lost, the restoration will have to pay a huge price, and it is even more difficult to "restore as before".

Through policy and market optimization direction, and the current problems mentioned above, such as the imperfect policy formulation system, the high degree of policy fragmentation, and the lack of specific measures, we can extract the following inspiration for policy implications:

First, speed up the introduction of intelligent regulatory technology policies. The technologies of big data, intelligence and so on should be fully used to create a new format of a "government + platform" supervision model, as well as to severely crack down on problematic platform companies with a lack of management, prominent safety hazards, serious illegal operations, and poor transportation service quality, and promote the development of the industry. Besides, the online ride-hailing platform should be urged to upload information and data to the "online ride-hailing supervision platform", so the data sharing of the online ride-hailing platform data and the network law enforcement system can be realized. The recorded data information and the actual operation data information can be used by the government to supervise the process. The training and introduction of government management personnel who are familiar with new technologies such as the Internet and big data should be strengthened, and a government regulation team that adapts to new formats of online ride-hailing should be created, so the application level of government regulation technology can be enhanced.

Second, promote the implementation of market advance and the retreat mechanism of "easy entry and strict control", and enhance the whole process supervision during and after the event. The online ride-hailing companies should be guided to: formulate inspection, assessment, and scoring standards for transportation safety, transportation pricing, operation management, service quality; refine assessment content, as well as the reward and punishment measures; perfect the service quality reputation assessment mechanism; enhance vehicle operation monitoring as well as online and offline service management capabilities of the platform company; and do well in the operation and management of vehicles and drivers. The construction of trade union organizations and associations in platform enterprises should be accelerated; the publicity system for negative lists of illegal behaviors in the industry should be established; the role of trade unions and associations in close contact with the people should be played; conflicts and disputes should be resolved in a timely manner; and the interests of all participants should be coordinated and handled intensively, especially for the care of online ride-hailing drivers.

Third, perfect service guarantee policies. The government market departments should be encouraged to provide door-to-door services, help and guidance should be given to the transformation, upgrading, and development of traditional transportation enterprises, and priority should be given to them to apply for permission to expand business. The policies related to the care of front-line grassroots drivers, in terms of residence permit points, legal aid for labor disputes, and low-interest vehicle leasing loans, should be perfected. Special parking spaces, parking lots, and passages for online ride-hailing in large-scale transportation hub areas, should be planned, constructed, and increased. The reception environment for drivers and stations should be perfected; convenience can be provided for both drivers and passengers to wait and pick up passengers. The strength of the media should be used to expand the channels of various media to make the propaganda of relevant policies.

## 6. Conclusions

Our study combines policy instruments with the TOE analysis framework to construct an innovative framework for analyzing the style of online ride-hailing policies, and to explore the current situation and future development direction of these policies in China. The results of our statistical analysis show that although the central-level policies of China on online ride-hailing have different styles at different stages of development, regulatory policies have the dominating position at each stage. The problems of singularity and fragmentation in the use of existing policy instruments require the government to improve them in the context of specific objectives of economic and social development. The above conclusions are of practical value for the coordination of the relationship of all interests, and the optimization of online ride-hailing policy.

**Supplementary Materials:** The following supporting information can be downloaded at: https://www.mdpi.com/article/10.3390/pr10102035/s1, Table S1. Textual information on relevant policy documents issued by the Chinese government from 2016 to 2022. Table S2. The process of coding policy texts.

**Author Contributions:** All authors contributed equally to this work. X.L. was responsible for reviewing and editing, validation, supervision, and investigation. S.Z. was responsible for reviewing and editing, conceptualization, visualization, software, and methodology. D.L. was responsible for validation and investigation. T.C. was responsible for supervision. Z.Z. was responsible for validation. All authors have read and agreed to the published version of the manuscript.

**Funding:** This research was funded by the Tianjin Social Science Foundation of China, grant no. TJGLQN20-001, China Postdoctoral Science Foundation, grant no. 2020M670636, Liberal Arts Development Foundation of Nankai University, grant no. ZB22BZ0332, and Fundamental Research Funds for the Central Universities of China, grant no. 63212079.

**Data Availability Statement:** The original contributions presented in the study are included in the supplementary materials.

**Conflicts of Interest:** The authors declare that the research was conducted in the absence of any commercial or financial relationships that could be construed as a potential conflict of interest.

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
