# Peer review of "Policy Evaluation and Policy Style Analysis of Ride-Hailing in China from the Perspective of Policy Instruments: The Introduction of a TOE Three-Dimensional Framework"

_processes, doi:10.3390/pr10102035_

Round 1

Reviewer 1 Report

many thanks to the editors for the invitation. I have read your work carefully.. Specific comments are as follows.

-The abstract should briefly describe the research background and policy implications. 

- Research gaps should be well mentioned in the introduction. A good research gap can give the reader more insight.

-why the method you have used is better than other methods ? how did you improve the method? 
-how can the government benefit economically from your research?

-your paper sounds like a review paper? please discuss the literature deeply

-please the discussion and conclusion can't be in one section, also please add policy recommendations, theoretical and managerial implications etc.
-Some of the most recent literature (last three years) should be considered and updated.

The following papers can be good examples to help you improve your paper:

-Qing, L.; Chun, D.; Ock, Y.-S.; Dagestani, A.A.; Ma, X. What Myths about Green Technology Innovation and Financial Performance's Relationship? A Bibliometric Analysis Review. Economies 2022, 10, 92. https://doi.org/10.3390/economies10040092

-Muhammad Sadiq, Fenghua Wen, Abd Alwahed Dagestani, Environmental footprint impacts of nuclear energy consumption: The role of environmental technology and globalization in ten largest ecological footprint countries,Nuclear Engineering and Technology,2022,https://doi.org/10.1016/j.net.2022.05.016.

-Qing, L.; Chun, D.; Dagestani, A.A.; Li, P. Does Proactive Green Technology Innovation Improve Financial Performance? Evidence from Listed Companies with Semiconductor Concepts Stock in China. Sustainability 2022, 14, 4600. https://doi.org/10.3390/su14084600

-Dagestani, A. A. (2022). An Analysis of the Impacts of COVID-19 and Freight Cost on Trade of the Economic Belt and the Maritime Silk Road. International Journal of Industrial Engineering & Production Research, 33(3), 1-16.
-Qing, L., Alwahed Dagestani, A., Shinwari, R., & Chun, D. (2022). Novel research methods to evaluate renewable energy and energy-related greenhouse gases: evidence from BRICS economies. Economic Research-Ekonomska Istraživanja, 1-17.
-You G, Gan S, Guo H, Dagestani AA. Public Opinion Spread and Guidance Strategy under COVID-19: A SIS Model Analysis. Axioms. 2022; 11(6):296. https://doi.org/10.3390/axioms11060296

"the manuscript still have a point not very clear , I hope they may explain ,what's your contribution to the theory ?you can choose the theory you think it fits better this article can be a good example could help you to understand what do I mean by "contribution to the theory" Eesley, C., Li, J. B., & Yang, D. (2016). Does institutional change in universities influence high-tech entrepreneurship? Evidence from China's Project 985. Organization Science, 27(2), 446-461. https://doi.org/10.1287/orsc.2015.1038
regards, go ahead

Author Response

Dear reviewer and editor:

On behalf of my co-authors, we thank you for your professional comments concerning our manuscript entitled “Policy Evaluation and Policy Style Analysis of Ride-Hailing in China from the Perspective of Policy Instruments: The Introduction of a TOE Three-Dimensional Framework” (ID: processes-1942947). 

We sincerely appreciate the reviewer’s comments. and we will keep on making efforts to do it better. We have checked the full text, and the errors pointed out have been revised, and the language, formatting and punctuation errors in other places have been revised together.

Once again, we appreciate for reviewer/editor’s warm work earnestly. If there are any further questions, please contact us without hesitation..

Looking forward to hearing from you.

Thank you and best regards.

Yours sincerely

Reviewer 3 Report

Dear Editor,

Thank you for giving me the opportunity to review the paper entitled "Policy Evaluation and Policy Style Analysis of Ride-Hailing in China

from the Perspective of Policy Instruments: The Introduction of a TOE Three-Dimensional Framework", with ID Number: 1942947.

I have completed reading the paper. This paper has been written in a good way. But, in this paper, the authors have not explained the managerial and economic implications of their presented method. These implications should be described as a new section.

Also, the English language of this paper is poor. There are several grammatical and syntax errors in the text.

So, I recommend major revising for this paper.

Yours sincerely,

Author Response

Response to reviewer and editor

Dear reviewer and editor:

On behalf of my co-authors, we thank you for your professional comments concerning our manuscript entitled “Policy Evaluation and Policy Style Analysis of Ride-Hailing in China from the Perspective of Policy Instruments: The Introduction of a TOE Three-Dimensional Framework” (ID: processes-1942947). These comments are all valuable and helpful for revising and improving our paper, as well as an important guiding significance to our researches. We have studied each comment carefully and made correction for your kind consideration. The main corrections in the paper and the responds to the reviewer’s and editor’s comments are as following:

1.Response to comment: In this paper, the authors have not explained the managerial and economic implications of their presented method. These implications should be described as a new section.

Response: We thank the reviewer for asking the question and absolutely agree with the comment. According to the reviewer's opinion, we have added one section to explained the managerial and economic implications of their presented method. And further adjusted the content related to the classification of policy instruments. For example:

1)In the last section of the introduction, we have discussed that:

This study bridges these research gaps by systematically examining the development process of China's central-level online ride-hailing policies and introducing a TOE framework to analyze the style of China's online ride-hailing policies based on policy classification by using policy instruments. There are two important reasons for using policy instruments for classifying these policies. First, according to the principle of moderation, things undergo an accumulation of quantitative changes before they undergo qualitative changes. The development of online car policies is also a process of quantitative to qualitative change that occurs under the influence of the external environment and internal factors. Therefore, the analysis of the style of online ride-hailing policy needs to include the dimension of time. In addition, policy formulation and implementation are directly influenced by policy objectives, where the policy instrument classification method plays a key role in the process of assessing the desired objectives. Therefore, this paper introduces the TOE framework to systematically examine the stylistic changes in China's central-level online ride-hailing policies. The combination of this framework and policy instruments provides a new theoretical perspective for policy analysis research, as it introduces the influence of environmental variables on policy based on the original classification of policy instruments considering internal factors, which can reflect policy styles in a three-dimensional way.

2)In the second and third paragraphs of Section 2.2, we have discussed that:

There is often no fixed way of classifying policy instruments, and researchers mostly used one type of policy instrument that is appropriate to the subject of study to classify policies. Some researchers categorised environmental policy instruments into market-based and command-based instruments, exploring how environmental policies should be designed. Others classified policy instruments that stimulate innovation into supply-push and demand-pull types and explored which policies can promote innovation in renewable energy technologies. However, recent studies tend to treat policy instruments as a single dimension of analysis when they are used to classify policies. In other words, the classification method using only policy instruments only examines the internal characteristics of the policy but ignores the external factor of the policy implementation environment, and thus is not conducive to a holistic view of policy development. Some researchers applied the technology-organization-environment (the following is abbreviated as "TOE") framework to policy analysis, examining the factors influencing the acceptance of digital transformation policies by advertising companies from three dimensions, addressing the problem of focusing too much on the policy itself and neglecting the influence of the environment in the process of policy analysis, inspiring this study.

2.Response to comment: The English language of this paper is poor. There are several grammatical and syntax errors in the text.

Response: We appreciate the reviewer’s suggestion apologize for the poor language of our manuscript. We have worked on the manuscript for a long time and the repeated addition and removal of repeatedly revised the sentences and sections expressions, which obviously led to poor readability. We have now worked on both language and readability and have also involved native English speakers for language corrections. And we thank International Science Editing ( http://www.internationalscienceediting.com ) for editing this manuscript. We carefully revised the article, corrected the grammatical problems. We will pay more attention to writing thesis in the future, learn more about English language and writing norms, and make language level of my own thesis higher and more standardized. For example:

1)In the last sentence of Section 2.1, we have revised the original sentence.

Sharif Naubahar and Xing Jack Linzhou compared the online ride-hailing policies formulated by Xi'an, Chengdu, Beijing and Guangzhou, and found that different local economic development environments, urban strategic positioning, the uniqueness of environmental protection and other factors associated with large differences in policymakers’ attitudes towards online ride-hailing, which has led to the formation of diversified local policy innovations [16].

2)In the first paragraph of Section 2.2, we have revised the original sentence.

Currently, the policy instrument is widely used in the construction of policy systems in various fields such as environmental protection [6-8], education [9-10], transportation [11], and diplomacy [12-13].

3)Other modified contents are marked in the red font.

In conclusion, we tried our best to revise and polish the manuscript point-by-point. Special thanks to you for all the comments! And we hope to learn more knowledge from you!

Once again, we appreciate for reviewer/editor’s warm work earnestly, and hope that the correction will meet with approval. If there are any further questions, please contact us without hesitation.

Looking forward to hearing from you.

Thank you and best regards.

Yours sincerely

Reviewer 4 Report

Abstract need to be rewritten by excluding or taking the concluding remarks/sentences to the conclusion section.

In introduction section, highlight the contribution of this article to show the strength of your work.

write a paragraph on the paper organization at the end of the introduction section.

"4. Policy Style Analysis of Online Ride-hailing Policy Based on Three-dimensional 276 Analytic Framework", require introductory sentence before going to 4.1.

Separate the discussion from the conclusion section and write the conclusion in a most precise manner possible.

Reference list is good.

Author Response

Dear reviewer and editor:

On behalf of my co-authors, we thank you for your professional comments concerning our manuscript entitled “Policy Evaluation and Policy Style Analysis of Ride-Hailing in China from the Perspective of Policy Instruments: The Introduction of a TOE Three-Dimensional Framework” (ID: processes-1942947). These comments are all valuable and helpful for revising and improving our paper, as well as an important guiding significance to our researches. We have studied each comment carefully and made correction for your kind consideration. The main corrections in the paper and the responds to the reviewer’s and editor’s comments are as following:

1.Response to comment: Abstract need to be rewritten by excluding or taking the concluding remarks/sentences to the conclusion section.

Response: This opinion is very important to us. We appreciate for your suggestions and questions. we rewrote the abstract, and we have deleted the conclusion section. For example:

Online ride-hailing in China not only brings convenience for the public, but it has caused several problems, such as inadequate supervision, data security risks, and financial risks. This new industry has also disrupted the traditional taxi market. China’s government implemented some policies which were initially disorderly tightening, and then formed the policy system responding to various needs for tackling these issues gradually. There are some policy fluctuations and regulatory effects during the period. Therefore, it is imminent to evaluate the online ride-hailing policy text. In this paper, we took 43 online ride-hailing policies as samples with the consideration of policy instruments and statistical inspection methods. In this paper, we constructed an innovative three-dimensional analysis framework by combining with content analysis, and further identify the ride-hailing policy development during different stages of development periods (2016-2022). Digging the problems existing in the new online ride-hailing policies were drawn by module division, unit coding, inductive statistics, quantitative evaluation of policy text content and TOE (technology-organization-environment) style analysis. Finally, we provide insightful policy recommendations for online ride-hailing policies, and committed to provide theoretical support and a decision-making basis for governance policies in the transportation industry.

2.Response to comment: In introduction section, highlight the contribution of this article to show the strength of your work.

Response: We absolutely agree with comments of the reviewer experts. We have rewritten the “Introduction” in response to your opinion. We refined the language and expression, that made these sections look more logical and readable. The highlight their contributions were discussed in the introduction and literature review. (For details, From the first paragraph on page 1 to the third paragraph on page 3, and such as colored fonts). In addition, we also talked about the contribution to government benefit economically. ( In the first paragraph of Section 5.2) For example:

1)In the second paragraph on page 3

These policies have gone through several different stages of development, presenting regulatory policies with different styles. However, policy-level research on the online ride-hailing industry has not received commensurate academic attention, with some notable exceptions, by empirically examining the acceptance of new policies by online ride-hailing drivers in Shenzhen, China. In particular, there is still a lack of scientific and systematic answers to the style and policy evaluation of China's central-level online ride-hailing policies.

2) In the first paragraph of Section 5.2

Declining in government authority and legitimacy has an impact on social stability, business environment, and policy implementation, which hinders local economic development. There is a law of reciprocity between government political gains and economic gains in online ride-hailing governance. The political impact on various interest groups caused by the waste of government political benefit, and the excessive consumption of government political benefit will endanger the legitimacy of the government and political security and stability. Once it is severely wastage, the restoration will have to pay a huge price, and it is even more difficult to “restore as before”. The above analyses are of great significance and practical value for the optimization of Chinese online ride-hailing policy, the coordination of the production and life rights of all interests in the online ride-hailing market, the scientific realization of the domination, application and allocation of political and economic resources, the solving local crises and maintaining the country’s political security and stability.

3.Response to comment: Write a paragraph on the paper organization at the end of the introduction section.

Response: We thank the reviewer for asking the question and absolutely agree with the comment. According to the reviewer’s opinion, we have added a paragraph to describe the “paper organization” ( In the third paragraph on page 3). And hoping that readers can clearly understand it. As follows:

First, according to the principle of moderation, things undergo an accumulation of quantitative changes before they undergo qualitative changes. The development of online car policies is also a process of quantitative to qualitative change that occurs under the influence of the external environment and internal factors. Therefore, the analysis of the style of online ride-hailing policy needs to include the dimension of time. In addition, policy formulation and implementation are directly influenced by policy objectives, where the policy instrument classification method plays a key role in the process of assessing the desired objectives. Therefore, this paper introduces the TOE framework to systematically examine the stylistic changes in China's central-level online ride-hailing policies. The combination of this framework and policy instruments provides a new theoretical perspective for policy analysis research, as it introduces the influence of environmental variables on policy based on the original classification of policy instruments considering internal factors, which can reflect policy styles in a three-dimensional way. Afterwards, basing on the view of policy tool and statistical test, online ride-hailing policies in recent years were used as samples, the three-dimension analysis framework by using content analysis method was established and module division, unit coding, induction and statistics, quantitative evaluation were applied. Then, we analyzed the online ride-hailing policy style in China and dug the problems existing from an analytical framework constructed from three dimensions: policy instruments, policy developments, and the TOE framework. Finally, some policy recommendations were proposed based on the issues and policy styles which were previously evaluated.

4.Response to comment: Policy Style Analysis of Online Ride-hailing Policy Based on Three-dimensional 276 Analytic Framework", require introductory sentence before going to 4.1.

Response: Thanks for the careful guidance of the reviewer. According to the reviewer's opinion, we have added one paragraph before going to 4.1, and we have described the contribution to the theory. (For details, the relevant contents are placed before Section 4.1) For example:

The formation of their own policy networks is realized by the multiple governance subject elements involved in the online car-hailing policy. At the core of the network structure, the government controls resource allocation, in which the regulatory authorities have some resources and the right to speak. What's more, the online car-hailing policy network and market relations have been influenced by the continuous development of digitalization and the external environment. In this case, the vertical and horizontal vulnerabilities of the system are highlighted, and the policy system with strong autonomy may make the system structure gradually develop towards the direction of unstable situation. This causes the main elements of the policy to undergo adaptive interaction and order breakthroughs in the allocation of resources as well as the acquisition and exercise of power. The original closed policy network structure should be overcome, the governance radius of the policy network should be expanded, and the three-dimensional policy tool and TOE framework evaluation should be adopt to truly make the public become the "destination" of the policy network. As a result, the theoretical framework of "institution-function" exists. The "resilience" of the policy network is kept by putting people first. This can help the formation of a localized online car-hailing policy network system with strong integration, high degree of system openness and interdependence of internal networks. In this section, we will analyse the online ride-hailing policy style in China from an analytical framework constructed from three dimensions: policy instruments, policy developments, and the TOE framework.

5.Response to comment: Separate the discussion from the conclusion section and write the conclusion in a most precise manner possible.

Response: We absolutely agree with comments of the reviewer experts! We appreciate for your suggestions and questions. We have modified this part, and have separated the conclusion from the discussion. In addition, we have added a part to describe the policy recommendations in the the Section 5.2. As follows:

1)From what has been discussed above, we may safely draw three conclusions:

- There is a need to improve the policy system of online ride-hailing by reasonably allocating the proportion of the three policy instruments corresponding to control, incentive and information transmission.

2)There is often no fixed way of classifying policy instruments, and researchers mostly used one type of policy instrument that is appropriate to the subject of study to classify policies. Some researchers categorised environmental policy instruments into market-based and command-based instruments, exploring how environmental policies should be designed. Others classified policy instruments that stimulate innovation into supply-push and demand-pull types and explored which policies can promote innovation in renewable energy technologies. However, recent studies tend to treat policy instruments as a single dimension of analysis when they are used to classify policies. In other words, the classification method using only policy instruments only examines the internal characteristics of the policy but ignores the external factor of the policy implementation environment, and thus is not conducive to a holistic view of policy development. Some researchers applied the technology-organization-environment (the following is abbreviated as "TOE") framework to policy analysis, examining the factors influencing the acceptance of digital transformation policies by advertising companies from three dimensions, addressing the problem of focusing too much on the policy itself and neglecting the influence of the environment in the process of policy analysis, inspiring this study.

In conclusion, we tried our best to revise and polish the manuscript point-by-point. Special thanks to you for all the comments! And we hope to learn more knowledge from you!

Once again, we appreciate for reviewer/editor’s warm work earnestly, and hope that the correction will meet with approval. If there are any further questions, please contact us without hesitation.

Looking forward to hearing from you.

Thank you and best regards.

Yours sincerely

Round 2

Reviewer 1 Report

The authors modified the paper with love and tried to improve every detail 
well done, bravo

Author Response

Response to reviewer and editor

Dear reviewer and editor:

We sincerely thank you for your professional comments, and we will keep on making efforts to do it better. Without your endless patience and support, this work would not have come to this point.

Once again, thanks again for all the support!

With best regards!

Yours sincerely

Reviewer 3 Report

The authors have addressed all my comments in this revision. So, I believe this revised version is now acceptable for publication in Processes.

Author Response

(The authors gave the same response as above.)

Reviewer 4 Report

The article is very well updated. Please, address the following remaining one.

Separate the discussion from the conclusion section and write the conclusion in the most precise manner possible.

Means "5. Conclusion and policy implications", change to "5. Discussion". And write an additional paragraph under 6. Conclusion with a maximum of eight sentences / 10 lines.

Author Response

Response to reviewer and editor

Dear reviewer and editor:

On behalf of my co-authors, we thank you for your professional comments concerning our manuscript entitled “Policy Evaluation and Policy Style Analysis of Ride-Hailing in China from the Perspective of Policy Instruments: The Introduction of a TOE Three-Dimensional Framework” (ID: processes-1942947). These comments are all valuable and helpful for revising and improving our paper, as well as an important guiding significance to our researches. We have studied the comment carefully and made correction for your kind consideration. The main correction in the paper and the responds to the reviewer’s and editor’s comments are as following:

5. Discussion

Based on our overall analysis of China's online ride-hailing policy, the following conclusions can be drawn: First, the current regulatory policy instruments account for the largest proportion of the online ride-hailing policy mix, and information transmission policy instruments are very lacking. Second, the policy development process of online ride-hailing can be roughly divided into three stages. Although the content of the policy portfolio at each stage is affected by multi-dimensional factors such as technology, organization, and environment, the overall policy style features have the characteristics of regulatory policy tools. Finally, the government's economic and social incentive policies for the development of the car-hailing industry have gradually increased, which is conducive to promoting the integrated development of new business formats and traditional industries. At the same time, car-hailing has certain advantages in the green sharing economy.

...

6. Conclusion

Our study combines policy instruments with the TOE analysis framework to construct an innovative framework for analyzing the style of online ride-hailing policies and to explore the current situation and future development direction of these policies in China. The results of our statistical analysis show that although the central-level policies of China on online ride-hailing have different styles at different stages of development, regulatory policies have the dominating position at each stage. The problems of singularity and fragmentation in the use of existing policy instruments require the government to improve them in the context of specific objectives of economic and social development. The above conclusions are of practical value for coordination of the relationship of all interests, and the optimization of online ride-hailing policy.

In conclusion, we sincerely thank you for your professional comments, and we will keep on making efforts to do it better. And we hope to learn more knowledge from you! If there are any further questions, please contact us without hesitation.

Looking forward to hearing from you.

Thank you and best regards.

Yours sincerely
